# Skin Color Retention in Red Potatoes during Long-Term Storage with Edible Coatings

**DOI:** 10.3390/foods10071531

**Published:** 2021-07-02

**Authors:** Esam Emragi, Sastry S. Jayanty

**Affiliations:** Department of Horticulture and Landscape Architecture, San Luis Valley Research Center, Colorado State University, 0249 East County Road 9N Center, Fort Collins, CO 81125, USA; esam.emragi@colostate.edu

**Keywords:** edible coatings, red skin potatoes, sensory characteristics, skin color, storage

## Abstract

In this study, we aimed to investigate the effect of edible coatings and cold storage conditions on the skin color of red potatoes (Ciklamen and Modoc) stored for six months at 4 ± 2 °C and 90 ± 5% relative humidity (RH). The four different formulations used were sodium alginate (F1), sodium alginate and potato starch (F2), zein and chitosan (F3), and chitosan, sodium alginate and potato starch (F4), in addition to the control treatment with distilled water. The treated samples were assessed periodically during six months of storage for changes in color, levels of reducing sugars, total phenolics and sensory qualities. The results indicated that the treatment with edible coatings significantly enhanced the chroma value of skin color, especially F1 and F2 formulations. However, these coatings instilled a limited effect on the level of reducing sugars. Moreover, F1 and F4 formulations exerted a significant effect (*p* < 0.05) on anthocyanin content examined after three months of storage. Alginate-based edible coatings significantly improved sensory evaluation, especially in terms of the color, gloss, and general acceptability of red skin potatoes.

## 1. Introduction

Potato (*Solanum tuberosum* L.) is one of the most consumed vegetables worldwide [1]. Appearance is crucial for marketing fresh potatoes. A shiny bright-colored skin, which is free of blemishes and flesh color, is an essential selection criterion for consumers. Smooth-skinned tubers, in particular reds, are vulnerable to skinning and easily noticeable. Red potatoes account for approximately 10% of the total potato production in the USA.

Red potatoes acquire skin color from the anthocyanin pigments present in the tuber periderm and peripheral cortex [2,3,4]. The red and purple potato cultivars possess the highest concentrations of phenolic and anthocyanin compounds, which exhibit antioxidant activity compared to the russet potato cultivars [5]. During tuber bulking, the color intensity and anthocyanin concentration in red skin potatoes are reduced due to an increase in tuber size [2].

Storage conditions greatly influence the physical and chemical properties of potatoes, as well as their nutritional value. The long-term storage requirements for potatoes are 90–95% relative humidity (RH) and an optimum temperature of 3.3–4.4 °C accompanied with appropriate sprout inhibitor applications. The indicators of good quality potatoes include firmness, texture, skin color appearance and no external defects or sprouts. Sometimes, consumers may reject the commodity if it has a faded or uncharacteristic color [6,7,8]. According to a North Dakota State University Extension article, the appearance and other quality-related issues can be accounted as one of the main reasons for the vendors rejecting 15% or more potatoes (Understanding and Managing Blemish Problems in Fresh Market Potato—Potato Extension (ndsu.edu)).

Consumers favor tubers with bright-colored skin, and growers strive to produce and maintain red skin color at harvest, storage and marketing [6,8]. As tubers mature in the field and in storage, the amount of anthocyanin in the periderm of red-skinned potatoes decreases [6,9]. Application of the auxin-type plant growth regulator 2,4-D (2,4-dichlorophenoxyacetic acid) was often applied to the foliage to enhance the skin color of red-skinned potatoes [10,11]. In earlier days, vegetable dyes were used to enhance the color, but negative feedback by consumers led to finding alternative methods to maintain the color in red skin potatoes [9]. Besides influencing the purchaser’s notion, color is also considered as an index of other quality features, such as flavor and nutrition [12].

Natural processes, such as transpiration and respiration, continue even after fruits and vegetables are harvested [13,14]. Edible films and coatings limit the exchange of water vapor, oxygen and carbon dioxide in fruits and vegetables. Coatings can be classified as food preservatives because of their ability to improve overall quality [15]. An edible coating may contain proteins, polysaccharides, lipids or a mixture of these compounds. Much research on edible materials has recently been focused on composite or multicomponent films to take advantage of each component individually and minimize their disadvantages [16]. A thin layer of edible coating can be directly applied to the food or commodity to form a primary envelope. Edible coatings have been used to preserve moisture and act as barriers against gaseous exchange. Overall, they improve the sensory and mechanical properties and prevent microbial infections, thereby extending the shelf life of the produce [17,18].

To the best of our knowledge, this is the first report on the use of edible coatings to maintain the skin color in red potatoes under cold storage conditions. In 2017, we conducted a preliminary study on Ciklamen and Modoc with zein, alginate and potato starch with methyl jasmonate (100 ppm), DPA (100 ppm) and chitosan (1.5%) (Appendix A). In this study, we aimed to determine the effect of different edible coatings on two types of red skin cultivars to extend the shelf life and maintain the quality of potatoes. We also investigated the outcomes of sensory, physical and nutritional properties upon treatment with these edible coatings.

## 2. Materials and Methods

### 2.1. Tuber Samples

This study was conducted during the 2018 growing season. We included two red-skin cultivars, Ciklamen and Modoc. Ciklamen is a European cultivar that is resistant to a variety of bacterial, fungal and viral diseases. It is a short oval fresh market potato with smooth, bright red skin and creamy-white flesh. Ciklamen has a high marketable yield, excellent taste and a high culinary profile. Modoc is a collaborative release by the Agricultural Experiment Stations of Oregon, North Dakota, California Idaho, and Washington. Modoc is round or oval in shape with white flesh and purplish-red skin. Freshly harvested tubers were obtained from the San Luis Valley Research Center. Tubers were harvested from mid–September to October 2018 and tubers were visually selected for uniformity in size, color and absence of blemishes and disease. Tubers were stored in a cold room at 4 °C with 95% humidity to reduce the field heat after harvest. Tubers were treated with coatings after the pulp temperature reached 4 °C. Twenty tubers were selected for each treatment. Before the coating was applied, tubers were washed with tap water and air-dried at ambient room temperature.

### 2.2. Coating Materials

Food-grade coating materials were used in this study. Zein, acid-soluble chitosan, potato starch, sodium alginate, Tween 20, acetic acid, ethanol and glycerol were purchased from Sigma-Aldrich, Inc. (St. Louis, MO, USA). Other materials used while preparing the coating formulations, such as corn oil, essential oils, cinnamon and oregano, were purchased from Walmart. Formulations were decided based on preliminary tests conducted in 2016 and 2017 (Appendix A).

### 2.3. Chemicals

The other chemicals, such as Folin Ciocalteu Reagent (FCR), sodium carbonate, gallic acid, AlCl3, quercetin hydrate, dinitro salicylic acid, crystalline phenol, sodium hydroxide, sodium sulfite, potassium sodium tartrate tetrahydrate, potassium chloride, sodium acetate, glucose and metaphosphoric acid were purchased from Sigma-Aldrich Corporation (St. Louis, MO, USA). All chemicals obtained were of analytical grade.

### 2.4. Preparation of Coating Solutions

The method described by Maftoonazad et al. [19] was used to prepare the sodium alginate coating (F1) solution with some modification. A 1.5% (*w*/*v*) of sodium alginate powder was dissolved in water by heating at 70 °C while stirring until the solution became clear. Glycerol was added (50% w/sodium alginate dry weight) as a plasticizer to the coating solution. Ascorbic acid was added at a concentration of 0.75 g/100 mL of the formulation. The mixture was homogenized for 5 min at 25,000 rpm using a Sorvall Omni-Mixer Homogenizer (Norwalk, CT, USA). A 2% (*w*/*v*) solution of calcium chloride was prepared and sprayed on the tubers coated with sodium alginate to induce the cross-linking of the coating film.

The other three composite formulations prepared were as follows. (1) Potato starch and alginate emulsion with oregano oil (F2). Potato starch (3% *w*/*v*) and sodium alginate (3% *w*/*v*) were prepared separately and then mixed at a ratio of 1:1. Glycerol was added (2% *v*/*v* formula solution) as a plasticizer. Oregano oil (0.75% *v*/*v* of formula solution) and Tween 20 (0.025 g/100 mL formulation) were added as an emulsifier agent. The mixture was homogenized for 5 min at 25,000 rpm using Sorvall Omni-Mixer Homogenizer (Norwalk, Conn. USA). (2) Zein and chitosan emulsion with the Oregano oil (F3), where the concentration in the formula was 4.5% zein and 1.5% chitosan. The acid-soluble chitosan coating solution was prepared by dissolving 1.5% (*w*/*v*) chitosan in 1% aqueous acetic acid. The mixture was homogenized for 2 min and shaken in a 60 °C water bath for 30 min, followed by cooling to room temperature. Zein (4.5% *w*/*v*) and chitosan (1.5% *w*/*v*) solutions were mixed together at a ratio of 3:1. Glycerol was added (5% *v*/*v* formula solution) as a plasticizer. Oregano oil (0.75% *v*/*v* of formula solution), and Tween 20 was added as an emulsifier agent. The mixture was homogenized for 5 min at 25,000 rpm. (3) Potato starch, chitosan, and sodium alginate with the Oregano oil (F4). Potato starch (0.5% *w*/*v*), chitosan (2% *w*/*v*), and sodium alginate (1% *w*/*v*) solutions were prepared separately and mixed at a ratio of 0.5:2:1. Glycerol was added (2% *v*/*v* formula solution) as a plasticizer. Oregano oil (0.75% *v*/*v* of formula solution), with Tween 20 was added as an emulsifier agent. The mixture was homogenized for 5 min at 25,000 rpm.

### 2.5. Application of Treatments and Storage

Tubers were divided into several sets based on the number of formulations. They were sprayed until all tubers were covered entirely with the coatings. The coating films on the surfaces of the tubers were then dried by blowing air for 15 min. The dried samples were packaged in plastic mesh bags. The treated tubers were stored at 4 ± 2 °C and 90 ± 5% relative humidity (RH) in the dark for six months.

### 2.6. Color Measurement

The colorimetric measurements of the tubers for each treatment during the storage were obtained through monitoring the color changes in the tubers at zero time, three, and six months using the method described by Manolopoulou et al. [20]. A Photovolt Instrument MiniScan Chromameter (Reston, VA, USA) was used to assess the color on CIE L^*^a^*^b^*^ chromatic space. The L^*^ variable is an indicator of the darkening or lightening of the color. The a^*^ scale measures the degree of red (+a^*^) color, while the b^*^ scale measures the degree of yellow (+b^*^) color. The instrument was calibrated using white and black standards. Fifteen tubers were selected from each treatment group. A flat spot was marked on each tuber to minimize the external light, such that it did not interfere with the measurements. The same spot was used each time to perform the color measurement. The chroma values of the tubers were calculated by using the chroma values equation. C* = (a*^2^ + b*^2^)^0.5^. The C* chroma value described the intensity of color in a sample.

### 2.7. Extraction Procedure for Assessing Total Phenolics, Flavonoids and Reducing Sugars

Potato tubers were collected from each treatment group at zero time, three and six months. After that, all tubers were cut into small pieces, frozen, freeze-dried and stored at −80 °C for further analysis. One gram of freeze-dried material was weighed in a 10 mL falcon tube and then 5 mL 95% methanol was added. The mixtures were vortexed for 1 min and the tubers were incubated overnight in an orbital shaker at 150 rpm and 25 °C. The homogenates were centrifuged at 5000 rpm for 25 min, followed by filtration using Whatman paper (40 nm). The remaining tubes were re-extracted under identical conditions. The final volume was made up to 10 mL using methanol and stored at −80 °C for further analysis. The previous extraction was used to determine the levels of total phenolics, flavonoids and reducing sugars.

### 2.8. Determination of Reducing Sugars

The levels of reducing sugars were determined using a previously described 96-well microplate assay with slight modifications [21]. At first, the dinitrosalicylic acid reagent was prepared (10 g/L dinitrosalicylic acid, 2 g/L crystalline phenol, 10 g/L sodium hydroxide, and 0.5 g/L fresh sodium sulfite). One hundred and twenty µL of the dinitrosalicylic acid reagent was added to each PCR tube (BioExpress, Kaysville, UT, USA), and 20 µL of the extract was added and mixed well. The mixture was heated in a water bath at 99 °C for 15 min, cooled at 4 °C for 1 min, and incubated at 20 °C to stop the further reaction. After thorough mixing of the contents in each tube, 100 µL of the mixture was transferred to a 96-well flat-bottom microplate with 40 µL of 400 g/L potassium sodium tartrate tetrahydrate solution. The plates were well mixed for 2 min in the plate reader, and the absorbance was measured at 570 nm. The glucose standard was prepared using 800 mL/L methanol, and reducing sugars were expressed as mg glucose per g of dry matter.

### 2.9. Total Anthocyanin Content

A previously described pH differential method by Madiwale et al. [1] was used with slight modifications to determine the total monomeric anthocyanin content. One gram of freeze-dried potato sample was homogenized in 5 mL of acidified ethanol (80%, with 0.1% *v*/*v* formic acid). The mixture was vortexed for 1 min every 15 min for 1 h. Then, 3 mL of chloroform was added and the tubes were vortexed every 10 min for 30 min. The tubes were then centrifuged at 5000 rpm for 20 min and stored overnight at 4 °C. The ethanol phase was collected and stored at 20 °C until further analysis. Ten microliters of the previous extract of each sample was added to 290 µL of potassium chloride (pH 1.0) and sodium acetate (pH 4.5). The absorbance was measured at 525 and 700 nm for both sets of solutions with pH 1.0 and 4.5. Total anthocyanin content was calculated using the following equation:(1)$$\mathrm{Total}\;\mathrm{monomeric}\;\mathrm{anthocyanin}\;\left(\frac{\mathrm{mg}}{L}\right)=\frac{\left(A\times \;\mathrm{MW}\times 1000\right)}{\varepsilon }\times 1.\;$$
where A =(A525− A700)pH 1.0−(A525− A700)pH 4.5, MW = 449.2 and ε = 26,900 are the molecular weight and molar absorptivity of cyaniding-3-glucoside, respectively, and 1 corresponds to the path length. Total anthocyanin content was presented as mg/g dry weight.

### 2.10. Sensory Evaluation

The sensory evaluation of the control and treated tubers for color, gloss, texture, odor, and overall acceptability was performed three months after the treatments and cold storage at 4 ± 2 °C and 90 ± 5% RH. A previously described method by Bai et al. [22] was used, with minor modifications. A panel of forty judges (22 women and 18 men) with ages ranging from 21 to 55 years were randomly selected. Panelists were asked to score for color, gloss, texture, odor, and overall acceptability where 1 represents extreme dislike, 2 moderate dislike, 3 neither like nor dislike, 4 moderate liking, and 5 extreme liking for color, gloss, texture, odors, and overall acceptability.

### 2.11. Experimental Design and Statistical Analyses

The data on color (*n* = 15) were statistically analyzed using repeated measures design, while the anthocyanins, reducing sugars, total phenolics, total flavonoids, vitamin C (*n* = 3), and sensory evaluation (*n* = 40) were analyzed using factorial design. All results were expressed as mean ± standard deviation (SD) values. The P values obtained using ANOVA for all responses are available in supplementary Appendix A. All data were subjected to analysis of variance (ANOVA) of potato cultivars (Ciklamen and Modoc) and storage times (zero, three and six months) separately for all coating formulations and control samples. Tukey’s test was used to compare differences between treatments during storage time for both cultivars, and treatments were significant considered at *p* < 0.05. All statistical analyses were performed using the R software, version 3.4.3.

## 3. Results

### 3.1. Color Change

The color changes in Ciklamen and Modoc during storage are presented in Figure 1. The effect of the edible coatings on the tubers was noticeable immediately after the treatment. The results of chroma value measurements revealed a significant (*p* < 0.05) effect of the treatment of tubers with edible coatings on color at all storage periods, including zero time, three and six months. F1 and F2 formulations were the most effective, whereas the F3 formulation was less effective. During the storage period there was no significant change in the skin color of the tubers after each treatment.

### 3.2. Reducing Sugars

The levels of reducing sugars in treated tubers stored at 4 ± 2 °C and 90 ± 5% RH for six months are shown in Figure 2. In the case of the Ciklamen cultivar, there was no significant difference in the levels of reducing sugars during the entire storage duration. However, there were significant differences (*p* < 0.05) in the case of Modoc. The F4 formulation decreased the concentration of reducing sugars to the highest degree, while F2 and F3 exerted a negligible effect. Meanwhile, the concentration of reducing sugars increased in all treatment groups relative to the storage time, possibly due to the impact of low temperature.

### 3.3. Total Anthocyanins

The total anthocyanin levels in treated tubers stored at 4 ± 2 °C and 90 ± 5% RH for six months are shown in Figure 3. In general, the effect of the treatments with different edible coatings was limited after three months of cold storage. The only exception was in the treatment of Ciklamen with the F4 formulation, where the increase in the level of total anthocyanins was significant. However, there was no significant effect (*p* > 0.05) of other edible coatings on the anthocyanin content in Ciklamen after six months of cold storage. In the case of Modoc, there was no significant difference in the anthocyanin content after three months of cold storage, while there was a significant effect after six months with the F4 formulation (*p* < 0.05). Meanwhile, the anthocyanin content decreased in most of the treatments relative to the storage duration. Occasionally, there was a slight decrease in the anthocyanin content observed after three months compared with that after six months.

Total phenolics, flavonoids, and Vitamin C levels were measured in all treated tubers along with controls at zero, three and six months of storage. The data are shown in the Appendix A.

### 3.4. Sensory Evaluation

Sensory evaluations were performed after three months of cold storage, and the results of the treated tubers are shown in Figure 4 and Figure 5. The F1 and F2 formulations were most effective in improving the sensory characteristics of the treated tubers, such as color, gloss, and general acceptability, compared to the control samples in both the cultivars. While the F3 formulation was not acceptable to the panelists, it was significantly counterproductive when compared to the control sample.

## 4. Discussion

A consumer’s first decision on quality is based on the visual appearance and color of the commodity. Appearance is one of the most important attributes that influences the consumers acceptability of any product [23,24]. From 2016 to 2018, we tested different formulations on three different potato cultivars (Rio Grande, Youkan Gold, and Purple Majesty) to study the effect of edible coatings on fresh potato tuber under different storage conditions. During this study, we noticed that the application of coatings had a positive effect on the skin color of the tubers. In 2016 we tested several formulas with simple compounds and at different concentrations. A preliminary study in 2016 showed that some materials are hard to apply (due to high viscosity and difficulty to dry). Formulations containing zein showed fungal growth but acted as good moisture barriers.

Our current studies suggest edible coatings, especially sodium alginate, enhance the color of red skin potatoes (Figure 1). There were significant differences (*p* < 0.05) in the chroma values between the control and treated tubers during cold storage. It is possible that these edible coatings provide a thick barrier against ethylene production and gaseous exchange between the tubers and environment, thereby delaying natural processes such as respiration and transpirational loss in coated tubers during cold storage. Additionally, high CO_2_ levels have been shown to decrease ethylene synthesis in tomatoes, which delays changes in color [25]. The presence of antioxidants (vitamin C and essential oil) in the formulation might be another reason for delaying the color changes during storage. The data on lightness (L*) revealed an insignificant gradual decrease during storage in both treated and controlled tubers (data not shown). However, there were significant differences in lightness (L*) between the coated samples and the control. The lowest L^*^ values were observed in the F1 and F2 treated groups, and the highest were in the control and the F3 treated groups. The presence of Zein in the formulation increased the L^*^ value, while alginate reduced it.

Anthocyanin concentrations have been shown to decrease during tuber growth in some red skin varieties [2] (Andersen et al., 2002). Lewis et al. [26] demonstrated the effect of temperature on red skin potato cultivars. They revealed that anthocyanin concentration decreased when the tubers were stored at 10 °C, but increased at 4 °C. Our results on anthocyanin estimation were consistent with the results obtained by Lewis et al. [26], as the anthocyanin content increased when the tubers were stored at 4 °C.

Chitosan-based coating were applied for different applications on potato tubers for reducing weight loss and maintaining firmness [27], resistance to fusarium dry rot [28], and also applied to reduce greening [29]. Chiabrando and Giacalone [30] reported that the application of a chitosan coating on blueberry delayed changes in anthocyanin content and antioxidant activity compared to the control. Moreover, as the duration of storage advanced, the levels of anthocyanin increased in both the treated and control blueberries. Furthermore, comparing the chitosan-treated blueberries to the other treatment groups, the increase in anthocyanin level was significantly higher [30].

Cortez et al. [31] reported that the processing and storage of strawberries without oxygen was a good method for anthocyanin production and color stability. An increase in the concentration of reducing sugars during cold storage due to starch conversion may have caused an increase in anthocyanin synthesis. Reducing sugars during cold storage have been shown to provide carbon skeletons for enhanced anthocyanin biosynthesis [2]. Eddy and Mapson [32] showed that when exogenous monosaccharides, such as glucose and fructose, were applied, the anthocyanin concentration in cress seedlings (*Lepidium sativum)* increased. However, they concluded that the effects of sugar were indirect.

The permeability of the natural skin of fruits and vegetables is essential for the respiration of living tissues. Consequently, through choosing the appropriate permeability of the coating film, gaseous exchange and respiration can be regulated to extend the product’s shelf life. The oxygen permeability (OP) of a film can be regulated using antioxidants such as citric acid (CA) or ascorbic acid (AA, as additives in the film composition. The selectivity of edible coating films towards O_2_, CO_2_ and water vapor leads to a delay in the natural ripening process [33]. The tubers treated with F1 and F2 formulations underwent delayed color change throughout the storage period.

Panelists evaluated the visual characteristics of coated tuber samples and assigned high scores to the F1 and F2 formulations in terms of color, gloss, texture and general acceptability (Figure 4). Since the evaluation was performed after three months of storage, the effects of edible coatings in delaying the loss of quality in terms of texture and color could be attributed to the inhibition of water diffusion. Martínez-Romero et al. [34] observed a similar phenomenon in sweet cherries, and a decrease in water diffusion was defined as the attributable reason. Li and Barth [35] suggested that the edible coatings reduce dehydration by physically limiting the air-filled surface tissue. The maintenance of texture in treated tubers (F1 and F2) may be attributed to their ability to prevent water loss [36].

The highest scores for visual characterization in terms of the gloss of red skin potatoes were assigned to the F1 and F2 formulations compared to the control after three months of cold storage. This is because F1 and F2 formulations imparted an attractive, natural-looking gloss to the treated red skin potatoes compared to other formulations and control. Additionally, the panelists assigned the lowest (*p* < 0.05) scores in terms of odor and general acceptability to the F3 treated samples. This might be due to the presence of zein in the formulation, which imparts whiteness and has a strong odor (Bai et al., 2003). Although the F3 coating induced negative effects on odor and general acceptability, it still maintained the quality characteristics of the treated tubers for the longest duration. This could be attributed to chitosan’s superior antioxidant and antimicrobial activities [37].

## 5. Conclusions

In this study, we demonstrated that the treatment of red skin potatoes with edible coatings (F1 and F2) increased their chroma values compared to the control. Edible coatings F3 and F4 increased anthocyanin content in both cultivars after three months of storage. The combination of potato starch, chitosan and alginate coating (F4) was more durable and retained tuber anthocyanin levels even after six months. The treatment with edible coatings significantly improved sensory evaluations, especially in terms of color, gloss and general acceptability of red skin potatoes. Alginate-based coatings enhanced the tuber red skin color (chroma value) and increased general acceptance based on sensory characteristics in both cultivars tested.

## Figures and Tables

**Figure 1 foods-10-01531-f001:**
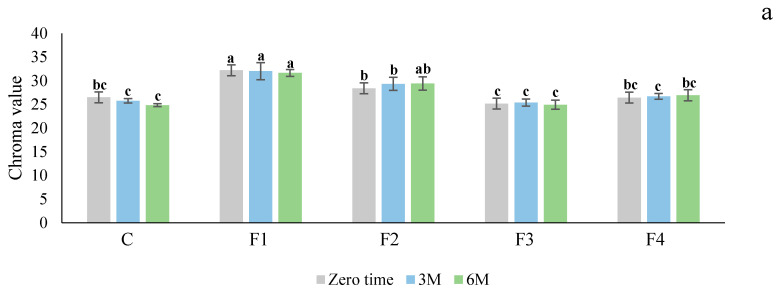
Effect of different edible coatings on the skin color of Ciklamen (**a**) and Modoc (**b**) after six months of cold storage at 4 ± 2 °C and 90 ± 5% RH. Data are expressed as mean ± SD, *n* = 15. The different letters on the bars are indicative of statistical significance (*p* < 0.05). C: control. F1: sodium alginate. F2: potato starch + sodium alginate emulsion with oregano oil. F3: zein + chitosan emulsion with oregano oil. F4: potato starch + chitosan + sodium alginate emulsion with oregano oil.

**Figure 2 foods-10-01531-f002:**
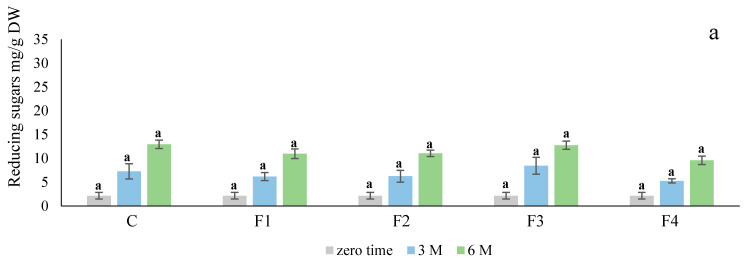
Effect of different edible coatings on the concentration of reducing sugars in Ciklamen (**a**) and Modoc (**b**) after six months of cold storage at 4 ± 2 °C and 90 ± 5% RH. Data are expressed as mean ± SD, *n* = 3. The different letters are letters on the bars are indicative of statistical significance (*p* < 0.05). C: control. F1: sodium alginate. F2: potato starch + sodium alginate emulsion with oregano oil. F3: zein + chitosan emulsion with oregano oil. F4: potato starch + chitosan + sodium alginate emulsion with oregano oil.

**Figure 3 foods-10-01531-f003:**
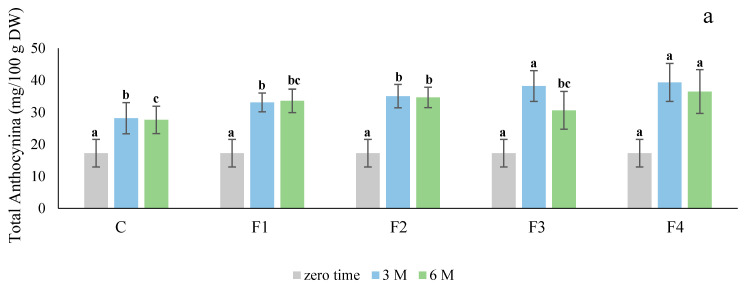
Effect of different edible coatings on total anthocyanin content of Ciklamen (**a**) and Modoc (**b**) after six months of cold storage at 4 ± 2 °C and 90 ± 5% RH. Data are expressed as mean ± SD, *n* = 3. The different letters are letters on the bars are indicative of statistical significance (*p* < 0.05). C: control. F1: sodium alginate. F2: potato starch + sodium alginate emulsion with oregano oil. F3: zein + chitosan emulsion with oregano oil. F4: potato starch + chitosan + sodium alginate emulsion with oregano oil.

**Figure 4 foods-10-01531-f004:**
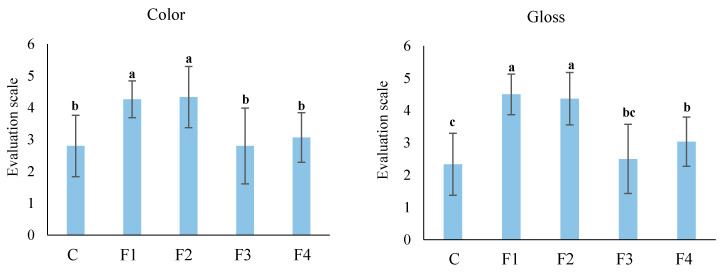
Effect of different edible coatings on sensory evaluations of Ciklamen and Modoc after six months of cold storage at 4 ± 2 °C and 90 ± 5% RH. Data are expressed as mean ± SD, *n* = 40. The different letters are letters on the bars are indicative of statistical significance (*p* < 0.05). C: control. F1: sodium alginate. F2: potato starch + sodium alginate emulsion with oregano oil. F3: zein + chitosan emulsion with oregano oil. F4: potato starch + chitosan + sodium alginate emulsion with oregano oil.

**Figure 5 foods-10-01531-f005:**
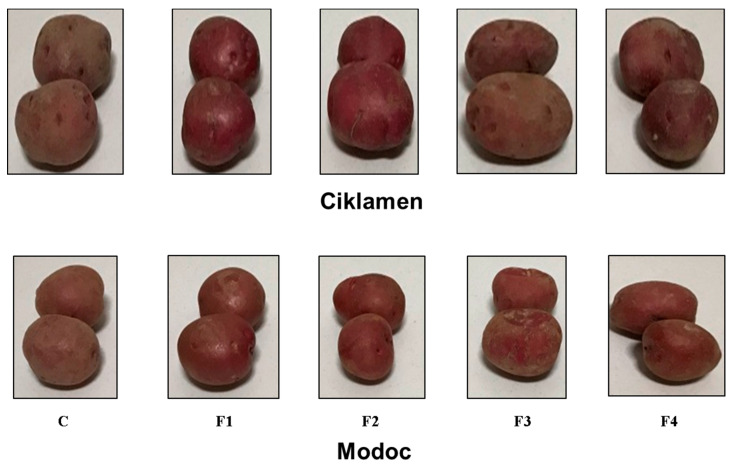
Effect of different edible coatings on the appearance of Cicklamen and Modac tubers after three months of cold storage at 4 ± 2 °C and 90 ± 5% RH. C: control. F1: sodium alginate. F2: potato starch + sodium alginate emulsion with oregano oil. F3: zein + chitosan emulsion with oregano oil. F4: potato starch + chitosan + sodium alginate emulsion with oregano oil.

## Data Availability

Not applicable.

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
