# Peer review of "Skin Color Retention in Red Potatoes during Long-Term Storage with Edible Coatings"

_foods, 2021, doi:10.3390/foods10071531_

Round 1

Reviewer 1 Report

This is an interesting study. It evaluated the effects of different coating formulations on skin color and other storage quality of two red potato cultivars during six-month storage. However I have some major concerns about the study:

1, this study is only based on one year of data collection, which is not sufficient to make solid conclusions. Potatoes used in the study were harvested from the field, and therefore the original quality (including skin set, chemical content) of those potatoes will be affected by growing conditions in the field. One year cannot cover year-to-year variation of tuber quality at harvest, and thus not enough for a solid storage study on potatoes. The reviewer strongly suggest that the authors repeat the study for at least one more year before considering a peer-reviewed publication. The current data is good for a research report published for extension purposes.

2. The manuscript lacked citations in its introduction section. Also a lot of the citations in the discussion section are irrelevant to potatoes or the Solanaceae family. The reviewer would suggest that the authors add more citations that are relevant to potatoes or the Solanaceae plants.

Specific comments:

Introduction: there are only 5 citations in the introduction part. Many of the statements need to be followed by a citation(s). Please add the citations. Otherwise those statements look subjective without scientific support.

Line 13: delete “period” after “storage”.

Line 20: please add a conclusion sentence at the end of the abstract.

Line 29: please add a citation here.

Line 35: delete “during bulking”

Line 38-39: please change it to “with appropriate sprout inhibitor applications”.

Line 40-41: please add a citation after “sometimes the consumers may reject the commodity if it has a faded or uncharacteristic color”.

Line 41-43: please add a citation after “…… for the vendors rejecting 15% or more potatoes”.

Line 45-46: please add citation after “…… vegetables are harvested”.

Line 50-51: please add citation after “…… to minimize their disadvantages”.

Line 54-55: please add citation after “… thereby extending the shelf life of the produce”.

Line 73: did you pick any particular tuber sizes for the experiment?

Line 88-105: please provide background on why those coating formulations were selected.

Line 119-121: please add citation after “…… the values of L*a*b were calculated using the special equations”.

Line 170: how were those 40 panelists selected? What are the gender ratio (men: women)? Age ratio? Need to provide more information on panelist selection.

Line 174-181: description of statistical analysis is vague. what is your experimental design? Please clearly indicate the fixed effects and random effects. Also I think your measurement during storage should be repeated measures, so clearly describe how ANOVA was run based on your experimental design.

Results section: please include a summary ANOVA table to show the p-values of main effects (I think it should be coating formulations, and cultivars) and their interaction for each measured trait. Otherwise it is unclear for the readers to understand how figures 1-4 are made.

Figure 1: The authors said “different letters on the bars are indicative of statistical significance”, but did not include any letters in Figure 1.

Figures 1-4: please double check on your letters. From high values to low values, it should be from “a” to “c” alphabetically. The letters in current figures are very confusing and don’t make sense.

Line 251-253: Could you please provide any evidence to show this hypothesis is correct? Otherwise it is just speculation, and it needs to be deleted. Or please add some citations that conducted similar studies.

Reviewer 2 Report

Very nice study. That said, I encourage the authors to better describe the following to improve manuscripts quality: 

Introduction: Add... What is done currently with the potatoes and how are they conventionally stored? What are the current losses (provide numbers for comparison purpose). The reader needs to have a better understanding of the current situation and why this is needed, and how these treatments can reduce losses and/or improve the quality and revenue.

Material and methods section

1) Better describe and cite quality measurements. For example, statement in lines 119-120 is vague: "After the basic 119 readings (x, y, and z) were taken, the values of L*a*b* were calculated using the special 120 equations". You need to specify: What equation, or at least cite it, is this common in this application and compared with others. Only include the values used in this study.  Apply the same to the remaining of the quality measurements, as all the others also appear to be vague.  2) In some cases, the study measured the same values through time (color, for example). Therefore, a 1-way ANOVA is not an appropriate methodology to compare this. Consider looking into a repeated measurement ANOVA. Authors need to significantly improve the description of the experimental design and analysis. Also, it is not clear what was measured by the panelist (all against all, or all against a reference), and how the authors reduced the dimension of the measurements to make sense of the data. Or was a simple two-way ANOVA performed per trait? Expand and augment.

3) Results, discussion and conclusion sections:

Recommendations are vague. The comparison is appropriate, but I suggest that the authors word a recommendation based on their findings, as it is obvious that some treatments have an effect. I am opposed to offering recipes, but I do believe that the authors need to offer more insight of the feasibility of application of new treatments.  In other words, if I am a farmer, what should I use now, in comparison to what I do, and what would this entitle. 

General Question: Can the authors specify if the evaluated treatments are approved for their application?  

Reviewer 3 Report

Detailed comments:

  • Lines 12-14: There are no results in the manuscript of the total chances in total phenols of potatoes.
  • Line 110: Correct the storage temperature from "40 °C" to "4 °C".
  • Lines 125 - 135: Section 2. Materials and methods describes "Extraction procedure for assessing total phenolics, flavonoids, and reducing sugars" The manuscript does not contain results for total phenolics and flavonoids of potatoes.
  • Lines 191 and 192 (Fig.1): Change the order in the legend to: Zero time, 3M, 6M.
  • Lines 194-196 and Fig.1: No SD on the bars and no letters on the bars, which indicative of statistical analysis.
  • Lines 201 - 203: The content of reducing sugars in the F4 group of the Modoc does not differ statistically significantly from the content of reducing sugars in the F1 group after 3 and 6 months (letters "a" and "ab", "abc"). So it should be: „F1 and F4 formulations decreased the concentration of reducing sugars to the highest degree”.
  • Lines 205 - 206: remove chart titles (top): "Concentration of reducing sugar in Ciklamen stored for six months at 4 °C and 90% RH" and "Concentration of reducing sugar in Modoc stored for six months at 4 °C and 90% RH ". Add "A" to the upper chart and "B" to the lower chart.
  • Lines 207 - 208: correct the title of Fig. 2 to: "Effect of different edible coatings on the concentration of reducing sugars in Ciklamen (A) and Modoc (B) after six months of cold storage at 4 ± 2 °C and 90 ± 5 % RH ". Add abbreviations to the formulation names, e.g. Alginate (F1), "Alginate and Potato Starch (F2) etc.
  • Lines 209-210: Correct the sentence: "The different letters are letters on the bars are indicative of statistical significance (P <0.05).
  • Lines 205 - 210 (Fig. 2): The reducing sugars contents are different for each potato group. Which reducing sugar values were compared to the Tukey's test? For example all green columns among themselves or for example all columns in the control group?
  • Lines 237 - 242: Whether the potatoes of Ciklamen and Modoc were tested together or separately. There are individual columns in the figures, not two columns for Ciklamen and Modoc separately.

Round 2

Reviewer 1 Report

Accept in current form

Reviewer 3 Report

All comments of the reviewer were included in the manuscript.